# Current Status of Mental Health in Mexico City

**DOI:** 10.3390/ijerph22081217

**Published:** 2025-08-02

**Authors:** Erika Sanchez, Humberto Nicolini, Jorge Villatoro, Marycarmen Bustos, María Elena Medina-Mora, Paola Mejía-Ortiz, Beatriz Robles, Diego Mondragón, Gema Ibarra, Daniela Meza, Alma Delia Genis-Mendoza

**Affiliations:** 1Facultad de Ciencia y Tecnología, Universidad Simón Bolivar, Mexico City 03920, Mexico; 2Laboratorio de Genómica de los Trastornos Psiquiátricos y Neurodegenerativos, Instituto Nacional de Medicina Genómica, Código Postal 14610, Mexico; hnicolini@inmegen.gob.mx; 3Instituto Nacional de Psiquiatría Ramon de la Fuente Muñiz, Tlalpan 14370, Mexico; ameth@inprf.gob.mx (J.V.); marybustos@inprf.gob.mx (M.B.); medinam@imp.edu.mx (M.E.M.-M.); paome.oz@gmail.com (P.M.-O.); 4Centro de Investigación en Salud Mental Global INPRFM UNAM, Mexico City 04510, Mexico; 5Universidad de la Salud, Álvaro Obregón 01210, Mexico; beatriz.robles@unisa.cdmx.gob.mx (B.R.); diego.mondragon@unisa.cdmx.gob.mx (D.M.); gema.ibarra@unisa.cdmx.gob.mx (G.I.); daniela.meza@unisa.cdmx.gob.mx (D.M.); 6Hospital Psiquiátrico Infantil “Dr. Juan N. Navarro”, National Institute of Genomic Medicine, Tlalpan 14080, Mexico

**Keywords:** mental health, Mexico City, prevalence, substance use, psychiatric disorders, epidemiological survey

## Abstract

There is limited information on the prevalence of mental health symptoms among the population of Mexico City. To provide an update and determine the prevalence of symptoms associated with various disorders in the city, a modified version of the “Screener Questionnaire” was used, the same instrument employed in the National Survey on Drug, Alcohol, and Tobacco Use (ENCODAT) 2016–2017. Data were collected at PILARES centers in different boroughs of Mexico City. A total of 868 questionnaires on symptoms of psychiatric disorders and the use of drugs, alcohol, and tobacco were completed. The most frequently reported symptoms were anxiety (50.92%), depression (38.02%), and post-traumatic stress disorder (43.09%). Additionally, results showed alcohol use at 15.1%, followed by tobacco (13.6%) and illicit drug use (6.8%). The prevalence of these symptoms was also compared with data from ENCODAT 2016–2017 to observe changes over the years.

## 1. Introduction

Mental health is a fundamental aspect of human well-being, significantly affecting both individual quality of life and social functioning. It plays a key role in daily coping, emotional regulation, and decision-making [1]. Mood disorders, anxiety, and substance use disorders are often associated with multiple risk factors, such as low educational attainment, violence, socioeconomic disadvantage, genetic vulnerability, and interpersonal difficulties [2].

Global studies, including those conducted in Mexico, have explored the lifetime prevalence of mental disorders through the World Health Organization’s World Mental Health Surveys, using *DSM-IV* diagnostic criteria. Prevalence rates vary widely across countries, with the United States reporting a lifetime prevalence of 47.7%, while Nigeria reports only 12.0%. Anxiety disorders are among the most common worldwide, with the highest rates found in the U.S. (31.0%) and Colombia (25.3%). Similarly, mood disorders show high prevalence in the U.S. (21.4%) and France (21.0%). Regarding substance use disorders, Ukraine (15%) and the U.S. (14.6%) present the highest figures [3].

To date, mental health has become a global priority, with more than 970 million people living with a mental disorder. These conditions account for between 13% and 15% of the global burden of disease and have increased significantly in recent decades, especially after the pandemic [4]. Depression, for example, affects more than 264 million people and is the leading cause of disability worldwide. Despite its high prevalence, the treatment gap remains alarming: between 76% and 85% of those in need in low- and middle-income countries do not receive adequate care [5].

In Mexico, the most recent nationwide mental health evaluation was conducted through ENCODAT 2016–2017, targeting individuals aged 12 to 65 through a randomized household survey. This study generated data on conditions such as mania/hypomania, psychosis, anxiety, depression, obsessive–compulsive disorder, PTSD, and suicide attempts, alongside information on alcohol, drug, and tobacco use [6].

The COVID-19 pandemic (2020–2023) introduced unprecedented challenges to daily life, disrupting routines and increasing psychological distress due to prolonged social isolation [7]. According to the WHO’s scientific brief “Mental Health and COVID-19: Early Evidence of the Pandemic’s Impact”, there was a significant global increase in mental health issues, particularly among young people and adults, including a rise in suicidal thoughts linked to loneliness and positive COVID-19 diagnoses [8].

In Mexico, these effects were especially pronounced among women, young individuals, and people working in informal employment. Despite global data showing an increase in psychiatric disorders in urban areas after the pandemic, there is still no specific data available for Mexico City. Updating this information is essential for developing public policies aimed at improving access to mental health care. Therefore, this study seeks to provide updated prevalence data on mental health symptoms and substance use across various boroughs of Mexico City, and to compare these results with those previously reported in ENCODAT 2016–2017 [9].

## 2. Materials and Methods

A cross-sectional, observational, and quantitative study was conducted with the aim of exploring the prevalence of various mental disorders and substance use in five boroughs of Mexico City.

Surveys were administered using an adapted instrument called “Screener Questionnaire” [9], which was linguistically and contextually adjusted for application in urban populations of the capital. The questionnaire included sections for identifying symptoms related to depression, anxiety, substance use, suicidal ideation, and other mental health indicators.

A simple random sampling technique was used, and the inclusion criteria were being 12 years of age or older and voluntarily agreeing to participate. The surveys were applied by nursing students from the Universidad de la Salud in different locations of the PILARES centers (Puntos de Innovación, Libertad, Arte, Educación y Saberes) across several boroughs of Mexico City, during the period between August and September 2023.

The data analysis was carried out in collaboration with the Ramón de la Fuente Muñiz National Institute of Psychiatry, with the goal of comparing the collected data with the results of the ENCODAT 2016–2017 survey, specifically those corresponding to Mexico City.

Regarding ethical considerations, all participants signed an informed consent form prior to completing the questionnaire. The study was reviewed and approved by the ethics committee of the Universidad de la Salud, and confidentiality and anonymity of the collected data were ensured at all times.

### 2.1. Instrument

The questionnaire used during the data collection included the following sections with the corresponding questions:

#### 2.1.1. Sociodemographic Data

Participants were asked to provide their name, sex, age, and date of birth.

They were asked if they were currently studying, having the following options: no, I have never been in school; no, but I went to school; and yes. This section also included the participant’s level of education.

Participants were asked about their marital status, having the following options of: married, divorced, domestic partnership, widow, separated, and single.

#### 2.1.2. Mental Health Information

Mania/Hypomania

Have you ever had a period lasting 3 or more days during which you felt unusually cheerful, irritable, energetic, or hyperactive—so much so that you felt or acted in a way that was clearly different from your usual self?Have you ever had a period lasting 3 or more days in which you needed very little (or no) sleep and still didn’t feel tired, or had more energy than usual?Have you ever been diagnosed with bipolar disorder (or manic-depressive disorder)?

Psychosis

4.Have you ever had a period during which you heard voices when no one was actually present, had visions, or saw things that others could not see?5.Have you ever had beliefs or ideas that others didn’t share with you, which you later discovered were not true?6.Have you ever been diagnosed with schizophrenia?

Anxiety

7.Have you ever had an experience where you suddenly felt very anxious or afraid and had physical symptoms of panic that got worse within 10 min?8.Have you had more than one such attack… and experienced a period of at least 1 month of intense worry about having another attack, or changed your behavior for at least 1 month because of the attacks?

Depression

9.Have you ever felt depressed, sad, down, or hopeless most of the day, nearly every day, for 2 weeks or more?10.Have you had a period of 2 weeks or more in which you lost most or all interest in your usual activities?11.During that period, did you also have feelings of worthlessness or guilt, or spend a lot of time thinking about death, suicide, or self-harm?12.During that time, did you notice a significant change in your appetite, an unintentional loss or gain of weight, changes in your normal sleep pattern, or trouble concentrating?

Post-Traumatic Stress

13.Have you ever experienced a traumatic event in which you felt your life was in danger, and had repeated images or strong memories of the event?

Suicidal Ideation

14.Have you tried to take your own life in the past 12 months?

Gambling

15.Do you frequently bet on races, fights, sports, or games?

Alcohol Use

16.Do you often drink more than 4 drinks in one day (for women) or more than 5 drinks in one day (for men)?

Tobacco Use

17.Have you ever smoked daily for a period of at least one month?18.Do you usually smoke your first cigarette within the first hour after waking up?

Drug Use

19.Have you ever tried to reduce or quit drug use and been unable to do so?

## 3. Results

The Table 1 presents the results obtained from the surveys conducted in 2023, with an *n* = 868 answers, where sociodemographic data such as age, educational level, marital status, internet use, and boroughs represented, and the psychiatric disorders surveyed this year, divided by men, women and the total.

The sociodemographic data show that most of the respondents were young people between 18 and 29 years (35.03% women and 35.61% men). The highest educational level was high school with 36.41%, with a similar distribution between genders. The most represented marital status was single (49.66% women and 56.47% men). In internet use, the principal response was more than 5 h per day (37.33% women and 35.61% men).

In psychiatric disorders, anxiety was the most reported disorder at 50.92%, depression at 38.02%, and obsessive–compulsive disorder at 30.65%.

This graphic shows the comparison between the prevalence of psychiatric symptomatology in Mexico City based on data from the ENCODAT 2016–2017 and the 2023 survey, broken down by sex (male and female).

The level of education was very similar across both surveys, although in 2023 the most common academic level among both men and women was high school, whereas in 2017 it was middle school. The proportion of individuals with undergraduate and postgraduate degrees increased in both genders by 2023.

Regarding marital status, the most frequently reported status in 2023 was single, which showed a significant increase compared to 2017, when the most common status was married or in a domestic partnership.

As for psychiatric disorders, it is worth noting that unlike in 2017—when the three most prevalent disorders were post-traumatic stress disorder, depression, and obsessive–compulsive disorder—the most recent data indicate that the top three in 2023 were anxiety, depression, and post-traumatic stress disorder.

## 4. Discussion

In the 2023 survey results, the most represented group was young people aged 18 to 29. This may influence the educational level results, with higher proportions in high school, undergraduate, or postgraduate education. Previous studies from INEGI [10] have found that higher education levels may be related to lower incidence of mental disorders. However, in our study, the most prevalent disorders were anxiety and depression, suggesting that other social and environmental factors might be involved.

One factor that may affect mental health is the area in which people live and carry out their daily activities. The most represented boroughs in our sample were Gustavo A. Madero and Álvaro Obregón, which include medium- and low-socioeconomic areas. This may influence access to health, psychological, and psychiatric services [11].

Another important factor for mental health is interpersonal relationships. Results show that single people reported a higher prevalence of mental disorders (51.84%) compared to those who were married or in a cohabiting relationship (35.17%). This aligns with other studies suggesting that marriage or stable relationships can provide emotional support [12]. However, the difference between people who are separated, widowed, or divorced and those who are married is small, which could mean that the quality of interpersonal relationships is more important than marital status itself for mental health.

The most prevalent disorders were anxiety (50.32%), followed by depression (32.73%) and OCD (28.86%). This trend is also seen in other studies, which show increases in anxiety and depression in recent years, especially after the COVID-19 pandemic [8,13]. Although increases were observed in both sexes, women showed slightly higher prevalence. A study by UNAM (2022) found that young women reported higher levels due to emotional burdens, family responsibilities, and greater exposure to gender-based violence. This may also explain the rise in PTSD cases [13].

Regarding substance use, illegal drug use has increased since the last survey [9], where 4% was reported in the general population and 17% in high-risk groups. This may reflect increased use among young people [6]. Tobacco use was lower compared to ENCODAT, possibly due to habit changes or the transition to vaping, which has increased among adolescents and young people [14]. Alcohol use showed no significant change and remained within similar parameters (25% in men and 8.5% in women), with higher prevalence in men [9].

In terms of education, a comparison between the 2017 and 2023 data showed an increase in high school and university levels, reflecting better access to education. This could be due to growing competition in the labor market, which increasingly demands higher education levels for well-paying jobs [15]. 

Marital status showed a significant increase in the proportion of single individuals in 2023 compared to 2017, which aligns with trends in recent generations to postpone marriage and prioritize financial and personal stability [16].

Data from Table 2 shows a significant increase in the prevalence of mental disorders between 2017 and 2023, especially in anxiety, psychosis, depression, OCD, and PTSD.

During the social isolation caused by the COVID-19 pandemic—with distancing, activity interruptions, and transitions to remote work, school, and socialization—anxiety and depression became more severe due to the abrupt change in daily routines, directly affecting mental health. The rise in family concerns, such as the loss of loved ones, economic uncertainties, and health issues, led to a general increase in mental disorders [17], as observed in Table 2.

Anxiety was the most prevalent disorder in 2023, affecting 55.0% of men and 51.7% of women, while it was barely reported in 2017. This increase is consistent with global studies reporting a 25% rise in anxiety and depression following the COVID-19 pandemic [7]. Research in Mexico [18] also showed increased levels of anxiety and PTSD among students and workers due to economic uncertainty and isolation.

Post-Traumatic Stress Disorder (PTSD) rose from 5.3% in 2017 to 44.5% in women and 45.0% in men in 2023. The impact of COVID-19, loss of loved ones, economic crises, and violence were key factors in this increase, according to studies by the UNAM Faculty of Psychology [18].

Psychosis increased from 3% in 2017 to more than 25% in 2023 across both genders. The Pan American Health Organization [19] has reported a rise in psychotic symptoms due to the use of psychoactive substances, which have also seen an increase, possibly due to extreme stress and precarious living conditions [20]. 

The data suggests that women reported higher levels of anxiety (51.7% vs. 55.0% in men) and depression (43.3% vs. 30.7% in men), while men had slightly higher prevalence of psychosis, OCD, and gambling disorder. These findings are consistent with international epidemiological studies showing that women are at greater risk of developing affective disorders, while men are more prone to impulse control disorders and psychosis [21].

It is important to emphasize that as mental disorders become more visible, people will feel more confident talking about them. This increased openness may also help explain the rise in reported cases in recent years. Despite this, the results for men may appear lower compared to women, as men face social pressures to appear strong and self-reliant, often striving for a hegemonic masculinity where seeking help is stigmatized. Shame, fear, and isolation lead many men to minimize their symptoms and avoid professional help, which can hide the true impact on men’s mental health [22].

Several studies have documented the global negative impact of the COVID-19 pandemic on mental health. According to the World Health Organization [23], anxiety and depression disorders increased by 25% worldwide due to social isolation, fear of infection, job loss, and disruption of health services. Similar to the Mexican context, research conducted in the United States, the United Kingdom, Canada, and Australia showed that young people aged 18 to 29 and women were the most affected groups, reporting higher levels of anxiety, depression, and substance use [24]. In many countries, there was also an increase in the use of alcohol, cannabis, and anxiolytics as coping mechanisms in response to lockdowns and uncertainty [25]. Furthermore, most countries reported partial or complete interruptions in mental health services during the most critical phases of the pandemic [23], which worsened the situation for vulnerable populations. Lastly, loneliness and isolation were key factors that intensified depressive symptoms, especially among young people, the elderly, and individuals living alone [26].

### Limitations

The sample size was small, which makes it difficult to generalize the results to the entire population. Moreover, since the data were collected through a self-report instrument, there is a risk of bias in the responses due to forgetfulness, subjective interpretation, or the stigma associated with mental disorders and substance use.

## 5. Conclusions

In recent years, the prevalence of mental disorders has increased significantly. To reduce this problem, it is essential to increase investment in research and establish monitoring systems that allow for the evaluation of trends in mental health and substance use. This will facilitate an accurate assessment of the national and local landscape.

Likewise, it is necessary to strengthen primary care by training medical and health care personnel in the early detection of mental disorders and addictions.

It is recommended to implement awareness and anti-stigma campaigns through both digital and traditional media, with special focus on young people, individuals with low educational attainment, and communities with limited internet access. In addition, it is crucial to design public policies tailored to the needs of each region, including the creation of community centers for comprehensive mental health care. These actions should ensure equitable access to mental health services, including psychological therapy, medications, and psychiatric follow-up through the public health care system.

## Figures and Tables

**Table 1 ijerph-22-01217-t001:** Surveys obtained in 2023.

	WOMEN		MEN			TOTAL	
	*n*	%	*n*	%	*p*	*n*	%
Age							
12–17	45	7.65	29	10.43	0.087	74	8.53
18–29	206	35.03	99	35.61		305	35.14
30–39	86	14.63	43	15.47		129	14.86
40–65	189	32.14	67	24.10		257	29.61
65 or more	61	10.37	39	14.03		101	11.64
Educational level		0.00					
Elementary school	57	9.69	24	8.63	0.742	81	9.33
Middle school	121	20.58	68	24.46		189	21.77
High school	214	36.39	102	36.69		316	36.41
Bachelor's degree or higher	132	22.45	66	23.74		199	22.93
Marital status		0.00					
Married/domestical partnership	208	35.37	97	34.89	0.006	305	35.14
Separated/widow	83	14.12	19	6.83		103	11.87
Single	292	49.66	157	56.47		450	51.84
Use of internet		0.00					
Frecuent	223	37.93	99	35.61	0.005	322	37.10
Regular	250	42.52	137	49.28		388	44.70
Low	41	6.97	8	2.88		49	5.65
Doesn´t use	57	9.69	27	9.71		84	9.68
Boroughs		0.00					
Iztacalco	78	13.27	19	6.83	0.009	97	11.18
Xochimilco	75	12.76	40	14.39		115	13.25
Gustavo A. Madero	126	21.43	45	16.19		171	19.70
Miguel Hidalgo	62	10.54	43	15.47		106	12.21
Alvaro Obregon	94	15.99	41	14.75		135	15.55
Mental health							
Mania/hypomania	39	6.63	21	7.55	0.722	60	6.91
Phsycosis	149	25.34	68	24.46	0.845	217	25.00
Anxiety	294	50.00	148	53.24	0.414	442	50.92
Depression	246	41.84	84	30.22	0.001	330	38.02
OCD	180	30.61	86	30.94	0.986	266	30.65
PTSD	251	42.69	123	44.24	0.720	374	43.09
Gambling	35	5.95	48	17.27	0.001	83	9.56
Suicide Attemp	32	5.44	9	3.24	0.209	41	4.72
Alcohol use	70	11.90	61	21.94	0.001	131	15.09
Drug use	30	5.10	29	10.43	0.005	59	6.80
Tobacco use	70	11.90	48	17.27	0.041	118	13.59

**Table 2 ijerph-22-01217-t002:** Comparison of prevalence percentages obtained in 2017 and 2023, from the CDMX divided into men and women.

	MEN					WOMEN				
	2023		2017			2023		2017		
	*n*	%	*n*	%	*p*	*n*	%	*n*	%	*p*
Educational level										
Elementary school	16	6.7%	7	9.7%	0.04273157	36	6.8%	27	21.1%	*p* < 0.001
Middle school	62	26.1%	31	43.1%		110	20.9%	43	33.6%	
High school	90	37.8%	20	27.8%		207	39.4%	37	28.9%	
Bachelor's degree or higher	61	25.6%	14	19.4%		128	24.3%	21	16.4%	
Marital Status										
Married/domestical partnership	73	31.1%	31	41.9%	*p* < 0.001	190	36.3%	66	50.4%	*p* < 0.001
Separated/widow	12	5.1%	7	9.5%		49	9.4%	24	18.3%	
Single	150	63.8%	36	48.6%		284	54.3%	41	31.3%	
Mental health										
Phsycosis	63	26.5%	2	2.7%	*p* < 0.001	132	25.1%	3	2.3%	*p* < 0.001
Anxiety	131	55.0%	0	0.0%	*p* < 0.001	272	51.7%	6	4.6%	*p* < 0.001
Depression	73	30.7%	5	6.8%	*p* < 0.001	228	43.3%	10	7.6%	*p* < 0.001
OCD	81	34.0%	3	4.1%	*p* < 0.001	165	31.4%	2	1.5%	*p* < 0.001
PTSD	107	45.0%	9	12.2%	*p* < 0.001	234	44.5%	7	5.3%	*p* < 0.001
Gambling	39	16.4%	11	14.9%	0.8570227	35	6.7%	4	3.1%	0.1481765
Suicide Attemp	9	3.8%	0	0.0%	0.1216412	31	5.9%	3	2.3%	0.1219529

## Data Availability

Data is unavailable due to privacy or ethical restrictions.

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
