# Peer review of "Current Status of Mental Health in Mexico City"

_ijerph, 2025, doi:10.3390/ijerph22081217_

Round 1

Reviewer 1 Report

Comments and Suggestions for Authors

The research is of interest and methodologically well-constructed and structured. It generally meets the requirements for publication.
In any case, some suggestions for improving the proposed work, which are primarily formal, are shared.
The length of the analysis is excessively brief and telegraphic in some sections. Specifically, a more detailed explanation of the subheading "2.1.2. Information on mental health," which has been presented in a purely schematic manner, would be appreciated. Even if it complies with the requirements of a maximum length article, a table or conceptual scheme that would bring together the categories in a more visual manner would be appreciated.
Likewise, the richness of the data reflected in the tables would allow for a broader discussion and the conclusions reached, which are equally concise.
On the other hand, regarding the final quote from the discussion section, "According to Dr. Jason Hunziker of the Huntsman Mental Health Institute, men face societal pressures to appear strong and self-sufficient. This can lead them to minimize their symptoms and seek professional help, which can obscure the true impact on men's mental health," the literal quote from which this statement is taken is not identified.
And regarding the conclusions, in relation to the statements: "The data suggest the need to strengthen mental health care and implement public policies that support these interventions" and "This will allow for more awareness campaigns in society and will help ensure access to mental health services and treatments," the quality and interest of the work would be greatly enhanced if some type of public policy proposal were developed in this regard: What policies do the researchers suggest? How can we proceed to strengthen health care? What types of awareness campaigns? How can we ensure access to services and treatments?
Finally, after reviewing the bibliography in open sources using the search engine "CURRENT STATE OF MENTAL HEALTH IN MEXICO CITY", publications of interest from 2024 and 2025 appear that are not reflected in the references used for the preparation of the research work, which, although it uses references and bibliography and is correct in terms of quality and relevance... should include the effort to be as up-to-date as possible due to the very essence of the subject analyzed and so that the work is as up-to-date as possible, reinforcing its interest.

Author Response

  •  Even if it complies with the requirements of a maximum length article, a table or conceptual scheme that would bring together the categories in a more visual manner would be appreciated.

I changed the way the criteria are presented.

  • "According to Dr. Jason Hunziker of the Huntsman Mental Health Institute, men face societal pressures to appear strong and self-sufficient. This can lead them to minimize their symptoms and seek professional help, which can obscure the true impact on men's mental health," the literal quote from which this statement is taken is not identified.

I revised that paragraph using a new citation, which is now properly identified.

  •  What policies do the researchers suggest? How can we proceed to strengthen health care? What types of awareness campaigns? How can we ensure access to services and treatments? 

I changed the conclusions to include what you mentioned.

Reviewer 2 Report

Comments and Suggestions for Authors

Thank you for the opportunity to review your work!  A couple of thoughts - In Materials and Methods - adding the survey sampling strategy (ie convenience, random etc) would be helpful;

Discussion - adding international comparisons if available - are your findings similar to what was observed in other countries regarding the impact of COVID?

Also - please add a strengths and limitations section to the discussion.

Small typos: Line 92 replace that with the - the last 3 days; Line 217 - Data show (instead of shows)

There were a few improvements I suggested - 1) Add the sampling strategy to the methods 2) For discussion include a review of findings related to mental health pre and post covid outside of Mexico if no comparable work there 3) Add a strengths and weakness section to discussion.

Author Response

Thank you for the review and the comments.

1) Add the sampling strategy to the methods

I revised the way I wrote the methodology section to make the strategy easier to understand.

2) For discussion include a review of findings related to mental health pre and post covid outside of Mexico if no comparable work there

I added a paragraph discussing how COVID affected mental health in other parts of Mexico.

3) Add a strengths and weakness section to discussion.

I added a paragraph next to the discussion

Reviewer 3 Report

Comments and Suggestions for Authors

Dear authors:
Your introduction is direct, precise and coherent, which is valuable. However, I believe it would be beneficial to strengthen it by including updated bibliographic references.

From lines 34 to 37, it would be useful to include references to support your assertions.
In the bibliography, you cite Ramos (2014), but this is not referenced in the document.
Please also include the meaning of the WHO acronym.
The final paragraph of the introduction lacks relevance as it does not include bibliographic citations for the statements made, particularly in lines 60–61.

Materials and methods:
As this is the most fundamental part of any research project, enabling replication in other contexts, it is important to include information on the type of study, ethical considerations, inclusion criteria, sampling technique, and the instrument used and how it was adapted. Similarly, the document does not include the statistics used. There is no evidence presented of the research procedure, such as how the data were collected, the duration of the study and general schedule, interventions (if any), and ethical aspects (informed consent, ethics committee approval and confidentiality).

I suggest writing the paragraph about sociodemographic data in the past tense.

Results:
The tables (Tables 1 and 2) are difficult to read, perhaps due to the font size or clarity.
In this section, I propose conducting further statistical analyses.

Author Response

Thank you for the review and the comments.

  1.  Materials and methods:
    As this is the most fundamental part of any research project, enabling replication in other contexts, it is important to include information on the type of study, ethical considerations, inclusion criteria, sampling technique, and the instrument used and how it was adapted.

I revised the way I wrote the methodology section to make the strategy easier to understand.

2. From lines 34 to 37, it would be useful to include references to support your assertions.
In the bibliography, you cite Ramos (2014), but this is not referenced in the document.
Please also include the meaning of the WHO acronym.
The final paragraph of the introduction lacks relevance as it does not include bibliographic citations for the statements made, particularly in lines 60–61.

All corrections you mentioned have been made.